# Parental-perceived home and neighborhood environmental correlates of accelerometer-measured physical activity among school-going children in Uganda

**Bernadette Nakabazzi**[1¤]*, **Lucy-Joy M. Wachira**[1☯], **Adewale L. Oyeyemi**[2☯],
**Ronald Ssenyonga**[3], **Vincent O. Onywera**[1☯]

**1** Department of Physical Education, Exercise and Sports Science, School of Public Health and Applied
Human Sciences, Kenyatta University, Nairobi, Kenya, **2** Department of Physiotherapy, College of Medical
Sciences, University of Maiduguri, Maiduguri, Nigeria, **3** Department of Epidemiology and Biostatistics,
College of Health Sciences, Makerere University, Kampala, Uganda

☯ These authors contributed equally to this work.
¤ Current address: Department of Biochemistry and Sports Science, College of Natural Science, Makerere
University, Kampala, Uganda
* bnakabazzi@gmail.com

Science Center at Houston School of Public Health,
UNITED STATES

**Data Availability Statement:** The authors confirm
that the data supporting the findings of this study

## Abstract

The benefits of physical activity (PA) on children's health and well-being are well estab-
lished. However, many children do not meet the PA recommendations, increasing their risk
of being overweight, obese, and non-communicable diseases. Environmental characteris-
tics of homes and neighborhoods may constrain a child's ability to engage in PA, but evi-
dence is needed to inform country-specific interventions in understudied low-income
countries. This study assessed the associations between parental-perceived home and
neighbourhood, built environment characteristics, and moderate-to-vigorous physical activi-
ty (MVPA) among children in Kampala city, Uganda. In this cross-sectional study, data
were obtained from 256 children (55.5% girls) aged between 10 and 12 years and their
parents. Children's MVPA was measured using waist-worn ActiGraph accelerometers. The
environments were assessed using a valid self-reported parent survey. Linear regression
models with standard errors (clusters) were used to analyze the relationship between envi-
ronmental variables and children's MVPA. Sex-specific relationships were assessed using
sex-stratified models. Play equipment at home ($\beta$ = -2.37, p <0.001; unexpected direction),
residential density ($\beta$ = 2.70, p<0.05), and crime safety ($\beta$ = -5.29, p <0.05; unexpected
direction) were associated with children's MVPA. The sex-specific analyses revealed more
inconsistent patterns of results with a higher perception of land use mix associated with less
MVPA in girls (irrespective of school type attended), and higher perceptions of sidewalk
infrastructure ($\beta$ = -12.01, p <0.05) and walking and cycling infrastructure ($\beta$ = -14.72, p
<0.05) associated with less MVPA in girls attending public schools only. A better perception
of crime safety was associated with less MVPA among boys and girls attending private
schools ($\beta$ = -3.80, p <0.05). Few environmental characteristics were related to children's
MVPA in Uganda, and findings were largely inconsistent, especially among girls. Future

are available within the article and its supplementary materials.

**Funding:** This research was funded by the African Development Bank-Higher Education in science and Technology (AfDB-HEST) and Makerere University Kampala, Uganda. The funders had no role in study design, data collection and analysis, decision to publish, or preparation of the manuscript.

**Competing interests:** The authors declare that there is no competing interest regarding the publication of this paper.

studies are needed to understand the ecological determinants of health-related PA behaviors among children in Uganda.

## Introduction

Childhood physical activity (PA) is a lifestyle behavior that protects against various non-communicable diseases (NCDs) and tracks through adolescence and into adulthood [1, 2]. However, recent global estimates reveal that more than 80% of school-going children do not engage in the recommended daily 60 minutes or more of moderate-to-vigorous physical activity (MVPA), putting them at risk of NCDs and premature mortality later in life [3]. Worldwide estimates showed that physical inactivity (PI) exceeded 53 billion dollars in direct health care costs in 2013 and was responsible for 3.2 million deaths [4]. Therefore, reducing PI is important to public health, as specified in the global efforts to reduce PI by 15% by 2030 [5]. Therefore, it is necessary to identify correlates of children's PA to guide informed intervention strategies and policies aimed at reducing PI among children [4]. PA is influenced by various factors at multiple levels, including individual, interpersonal, environmental, and policy [5]. Although there is evidence on the importance of individual characteristics (e.g., age, sex, weight status) on children's PA [3, 4, 6–11], such socio-demographic factors cannot fully explain the insufficient PA exhibited by the majority of children. Interventions based on such evidence may only benefit a few individuals who deliberately want to be active and have generated limited effects [12].

Socio-ecological models recognize that PA behavior is influenced by the environment in which it occurs [7]. Home and neighborhood environments are two components of the PA environment relevant to children's active lifestyle. The home environment plays a major role in shaping and promoting children's PA through social support, enforcement of rules for PA, and creating a healthy home environment that supports PA through the provision of play equipment and limiting access to media equipment [13–15]. Parents and guardians play a vital role in influencing children's PA because they arrange home space and determine the equipment that is available for children's use [15]. Parents can support their children's PA by encouraging, facilitating, modeling, spectating/supervision, and co-participation [14, 16, 17]. Rules for PA are also a means through which parents control the home environment; parental constraints in the form of rules may lower children's MVPA [14]. In addition, playing and electronic and media equipment at home have been considered as important correlates of children's PA [14, 15, 18–21].

The neighborhood environment includes all places built or designed by humans, such as buildings, roads, recreation facilities, walking/cycling infrastructure, and urban design, as well as fewer tangible factors, such as safety [22]. PA-friendly neighborhoods make it easy, accessible, safe, and comfortable for children to be active [23]. Review studies have concluded that the presence of recreational facilities, such as parks, playgrounds [24–28], and neighborhood walking/cycling infrastructure [27], are positively correlated with children's PA. In a systematic review and meta-analysis on street connectivity and PA, Jia et al. found that higher street connectivity predicted higher levels of MVPA in children [29]. Moreover, low street connectivity (cul-de-sacs) and the esthetic quality of neighborhood surroundings have been associated with children's PA [30]. A higher perception of community safety is also correlated with higher PA [31]. Nonetheless, parental concern about neighborhood safety (crime, traffic, personal, stranger danger) may lead to parents restricting their children's use of active travel

modes to neighborhood destinations, including school, outdoor play, and independent mobility, thus reducing their PA [14, 23].

However, this evidence has been largely informed by studies from high-income countries (HICs); translating findings from these studies to Sub-Saharan Africa (SSA) where the built environment and culture are different may be a challenge [6, 27, 28, 32]. Very few studies among school-going children have been conducted in SSA, and the results of these studies are inconsistent. For example, among Kenyan children aged 9 to 11 years, parental perceptions of positive neighborhood social interactions, safety, and connectivity were associated with MVPA compliance [33]. While in South Africa, among 9-to 10-year-old children, parental perceptions of the neighborhood environment were not related to children's MVPA [34]. Kampala is transitioning to a vibrant, attractive, and sustainable city through retrofitting and expansion of the city to accommodate the rapidly growing urban population [35]. Exploring the associations between children's PA and parental perceptions of the home and neighborhood environments in Kampala, Uganda is a timely issue because the results of the current study may inform the city's on-going re-design.

Exploring the associations between children's accelerometer-measured PA and parental perceptions of the home and neighborhood environmental attributes in Uganda may address the problem of quality PA data noted in many low-income communities (LICs) and provide country-specific evidence that can inform effective interventions and policies. Therefore, the aim of this study was to explore the relationships between the parental-perceived home and neighborhood environment with accelerometer-measured MVPA among school-going children in Uganda.

## Materials and methods

### Design and participants' recruitment

This cross-sectional study used a multistage sampling method to recruit a gender-balanced sample of 600 children aged between 10 and 12 years from mixed-day primary schools across Uganda's capital city, Kampala. Kampala comprises five administrative divisions (Central, Nakawa, Rubaga, Kawempe, and Makindye) covering an area of 189 km$^2$ and a population of 1.5 million inhabitants [36]. The Uganda National Council of Science and Technology (SS 4340) and Kenyatta University Ethical Review Board (PKU/619/703) granted ethical approval for this study. Divisions were the primary sampling unit; two divisions, namely central and Nakawa, were randomly selected. The second sampling unit was schools. Because of the variability in socioeconomic status (SES) between schools in Kampala (public schools represent the lower socioeconomic strata, and private schools represent the high socioeconomic strata based on the fee structure), four private and three public schools were selected for public and private school attendance. The third sampling unit was classes in the selected schools that best corresponded to 10- to 12-year-old children. At least 70 to 100 children in grades 5–7 were selected from each school. A total of 600 child-parent pairs received survey packages that contained parent and child information sheets, consent and assent forms, and a parental questionnaire. Written informed parental consent and child assent were obtained from all participants. To be eligible, children had to be 10 to 12 years old, residents of Kampala, and without physical disability. Children attending boarding schools were not included. Only 256 child-parent pairs provided complete consent and the data required for the study (response rate, 42.6%). We measured the children's MVPA using an accelerometer. Parents completed a survey that assessed home and neighborhood environment attributes and sociodemographics. The survey was adapted from the Neighborhood Impact on Kids study [18] and the Neighborhood Environment Walkability Scale for Africa (NEWS-Africa) [37] (see S1 File).

## Measures

**Children's physical activity.**   Children wore the ActiGraph GT3X+ accelerometer (Acti-Graph LLC, Pensacola, Florida, USA) around their waist for seven consecutive days, including two weekend days for 24 hours. Data were collected at a sample rate of 80 Hz with an epoch of 1 s and then aggregated into 15s epochs. Accelerometer data were processed using ActiLife software (Version 6.13.3). The Sadeh algorithm was used to identify sleep time and determine the wake-wear time [38]. Non-wear time within a day was set at 20 consecutive minutes of 0 counts. The analysis was restricted to data from children who had at least four days, including one weekend day, with ten or more waking wear hours each day. Evenson's age-specific cut points ($\geq$574 counts/15 s) were used to generate the average daily minutes of MVPA [39].

**Home environment.**   The home environment was assessed using survey items completed by the parent on the presence of play equipment, child's bedroom and personal media equipment, parental rules for PA, and parental social support [18]. The items demonstrated good-to-excellent test-retest reliability (ICC = 0.51–0.96) [40]. Parents reported support for their child's PA from a series of questions, including how often they encouraged children to be active, provided transport to places to do PA or sport, watched children engage in PA, and participated in PA with their child. The questions were rated on a 5-point Likert scale ranging from "none" to "daily." The four items were averaged to create the social support scale used for the analysis. The parental rules for PA score summed "yes" responses on 15 rules that elicited a Yes/No response from parents. For example, "stay close/within sight of the house/parent" to "Respect others (particularly adults)." Parents reported whether their child had the following play equipment at home: bike, basketball hoop, jump rope, active video game (e.g., Wii) sports equipment (e.g., balls, racquets, bats, and sticks), skateboard/roller skaters/scooter, fixed play equipment (e.g., swings), home aerobic equipment (e.g., treadmill, cycle, cross-trainer), weigh-tlifting equipment (e.g., free weights, exercise balls), yoga/exercise mat, recreational room, trampoline, and stairs. A play equipment score was generated using the sum of all the individual play equipment (range, 0 to 13). Parents also reported whether their child's bedroom contained the following media and electronic items: TV, computer, VCR or DVD player, and video game system (e.g., x-box, play station). A score was generated using the sum of electronic and media items in the child's bedroom (range 0 to 4). Parents/guardians further reported whether the child had personal media and electronic items, such as a cell phone/2-way radio, handheld video game players (e.g., Sony psp), and/or a music player (e.g., MP3, I Pod). The sum of the child's personal media and electronic equipment was generated (range, 0 to 3).

**Neighborhood environment.**   The NEWS-Africa [36] was used to assess parents' perception of their neighborhood environment attributes that may support children's PA. NEWS-Africa was adapted to the African context to help build on neighborhood-built environment research, and the 76-item instrument demonstrated excellent (ICCs > .75%) or good (ICCs = 0.60 to 0.74) test-retest reliability [36]. The NEWS-Africa is organized into 14 sub-scales representing residential density (1 item), land use mix-diversity {destinations} (21 items), land use mix-diversity {recreation} (4 items), land use mix access (7 items), street connectivity (5 items), side walk infrastructure (5 items), path infrastructure (2 items), crossing infrastructure (4 items), overall walking/cycling infrastructure (12 items), esthetics (8 items), traffic safety (6 items), crime safety (4 items), personal safety (3 items), and stranger danger (3 items). All the NEWS-Africa scales, except residential density and land use mix-diversity (destination and recreation), were rated on a 4-point Likert scale ranging from "strongly agree" to "strongly disagree." The residential density scale was rated on a single unweighted scale. Land use mix diversity (destination and recreation) was rated on a 5-point scale (i.e., 1–5 min, 6–10 min, 11–20 min, 21–30 min, and 31+ min). Items were scored as recommended by the

NEWS-Africa developers and reverse-coded where necessary [36]. For analysis, all the neighborhood environment attribute scores were averaged within each scale, and higher scores were expected to be associated with more MVPA.

**Covariates.** Parents reported their child's date of birth, sex, and time spent at the current residence.

## Data analysis

Data analysis was restricted to children who had complete parent surveys and accelerometer data (n = 256). We computed descriptive statistics, including percentages, means, and standard deviations, for children's characteristics and PA. Independent t-tests were used to compare parental perceptions of home and neighborhood environment variables by children's sex, school type, and compliance with PA guidelines. Logistic regression analyses with robust standard errors (clusters) were used to examine the associations between parent-perceived home and neighborhood environment variables and children's MVPA. Overall and sex-stratified logistic regression analyses were conducted. STATA (v.14.2, StataCorp, Texas, USA) was used for all analyses, and significance was set at $p \leq 0.05$.

# Results

## Participant characteristics

The study sample included 256 children. The majority of the children were between 10 and 11 years old (71.5%), 58.6% attended private school, and 55.9% were girls. The children had lived in their current address for an average of 6.1 ± 3.3 years and spent an average of 56 ± 25.7 minutes/day in MVPA. Among the study participants, the boys (60.1 ± 28.2) spent significantly more time in MVPA than the girls did (58.2 ± 23.0). Public school (low SES schools) children engaged in 26 minutes more MVPA than private school (high SES schools) children. No significant differences in children's characteristics were observed between those included and excluded from the analysis. The results are shown in Table 1.

## Home and neighborhood environment attributes and children's characteristics (MVPA compliance, sex, and school type)

Table 2 shows the mean differences in perceived parental home and neighborhood environment attributes according to children's MVPA categories. Parents of children who did not meet the MVPA recommendations reported more play equipment at home (p <0.001) and higher perception of neighborhood crime safety (p = 0.022) but lower perception of residential density (p = 0.012) than their counterparts whose children met the MVPA recommendations. Differences in perceived parental home and neighborhood environment attributes by school type and sex are also provided in Table 2. Parents whose children attended private schools (high SES) reported significantly more rules for PA (p = 0.033), children's personal media equipment (p = 0.002) and play equipment at home (p <0.001) than parents whose children attended public schools (low SES). Attributes of the neighborhood environment also varied across school types and children's sex. Parents of children in private (high SES) schools perceived higher crime safety relative to parents of children in public (low SES) schools (p = 0.012). Children in public (low SES) schools had parents who perceived a higher residential density (p <0.001) and street connectivity (p = 0.024) compared to parents of children in private (high SES) schools. Parents of girls perceived a significantly higher residential density compared to parents of boys (p = 0.024).

**Table 1. Children's characteristics and descriptive statistics (n = 256).**

| Variable | n | % |
|---|---|---|
| **School type** | | |
| Private (high SES) | 150 | 58.5 |
| Public (low SES) | 106 | 41.4 |
| **Sex** | | |
| Male | 113 | 44.1 |
| Female | 143 | 55.9 |
| **Age (years)** | | |
| 10 | 88 | 34.8 |
| 11 | 94 | 36.7 |
| 12 | 74 | 28.5 |
| Time spent at the current address (Mean ±SD) | 6.1 ± 3.3 | |
| **Average MVPA (minutes/day)** | | |
| | **Mean ± SD** | **p-Value** |
| **Overall** | 56 ± 25.7 | |
| **Sex** | | |
| Female | 52.8 ± 23.0 | 0.023* |
| Male | 60.1 ± 28.2 | |
| **School Type** | | |
| Private (HSES) | 45.4 ± 17.8 | <0.001** |
| Public (LSES) | 71.2 ± 27.5 | |
| **Age** | | |
| 10 | 52.7 ± 20.9 | 0.268 |
| 11 | 56.5 ± 26.5 | |
| 12 | 59.6 ± 29.4 | |

SD, standard deviation; MVPA, moderate-to-vigorous physical activity; BMI, body mass index; HSES, socioeconomic status;

*p <0.05,

**p <0.001.

## Parental-perceived home and neighborhood-built environment correlates of children's MVPA

In the overall model of the entire sample, at the home level, results showed that children spent less time in MVPA if their parents reported more play equipment at home ($\beta$ = -2.37, p <0.001). At the neighborhood level, significant positive associations were found between children's MVPA and parental perceptions of high residential density ($\beta$ = 2.70, p <0.05). Conversely, negative associations were found between children's MVPA and parental perceptions of high crime safety ($\beta$ = -5.29, p <0.05).

School-stratified models were created to identify correlates that may be unique to private (high SES) and public (low SES) school children (Table 3). The results of this study indicated that none of the home environment attributes were correlated with children's MVPA in both private and public schools. At the neighborhood environment level, a higher perception of crime safety was associated with less MVPA among children in private (high SES) school ($\beta$ = -3.80, p <0.05). Table 3 also presents the sex-stratified models. Parental perception of higher land use mix accessibility was associated with less MVPA among girls, regardless of school type (school SES). Our results further revealed that for girls who attended public schools (low

**Table 2. Mean differences in parental perceptions of the home and neighborhood-built environment attributes across children's school type (school SES), sex and MVPA compliance.**

| Variable | School type Mean (SD) | | P-value | Sex Mean (SD) | | P-value | MVPA Mean (SD) | | P- value |
|---|---|---|---|---|---|---|---|---|---|
| | Private (HSES) | Public (LSES) | | Male | Female | | Sufficient PA | Insufficient PA | |
| Home Level | | | | | | | | | |
| Parental support for PA | 2.7 (0.9) | 2.6 (1.1) | 0.411 | 2.6 (0.9) | 2.6 (1.0) | 0.725 | 2.5(1.0) | 2.7(1.0) | 0.121 |
| Parental rules for PA | 13.2(1.9) | 12.7(1.9) | 0.033* | 13.1 (1.7) | 12.9 (2.1) | 0.235 | 12.9(1.7) | 13.0(2.0) | 0.479 |
| Media equipment in child's bedroom | 0.5 (1.0) | 0.5 (0.9) | 0.756 | 0.5 (1.0) | 0.5 (0.9) | 0.575 | 0.5(1.0) | 0.5(0.9) | 0.995 |
| Child's personal media equipment | 0.7 (0.9) | 0.4 (0.7) | 0.002* | 0.6 (0.9) | 0.5 (0.8) | 0.203 | 0.4(0.8) | 0.6(0.9) | 0.055 |
| Play equipment at home | 4.8 (2.5) | 2.7 (2.1) | <0.001** | 4.2(2.5) | 3.6 (2.6) | 0.062 | 3.1(2.3) | 4.4(2.6) | <0.001** |
| Neighborhood Level | | | | | | | | | |
| Residential density | 2.4 (1.2) | 3.8 (2.0) | <0.001** | 2.7 (1.6) | 3.2 (1.8) | 0.024* | 3.4(1.9) | 2.8(1.6) | 0.012* |
| Land use mix-diversity (destinations) | 2.7 (0.7) | 2.6 (0.8) | 0.176 | 2.6 (0.7) | 2.7 (0.7) | 0.413 | 2.7(0.8) | 2.7(0.7) | 0.618 |
| Land use mix-diversity (recreation) | 2.0 (1.0) | 2.0 (1.1) | 0.695 | 2.1 (1.0) | 2.0 (1.0) | 0.395 | 2.0(1.0) | 2.0(1.0) | 0.720 |
| Land use mix-access | 2.9 (0.6) | 2.9 (0.6) | 0.698 | 2.9 (0.6) | 2.9 (0.6) | 0.710 | 2.8(0.6) | 2.9(0.6) | 0.440 |
| Street connectivity | 2.8 (0.5) | 3.0 (0.6) | 0.024* | 2.9 (0.5) | 2.9 (0.6) | 0.834 | 2.9(0.6) | 2.8(0.6) | 0.297 |
| Sidewalks infrastructure | 2.3 (0.7) | 2.2 (0.7) | 0.077 | 2.3 (0.7) | 2.2 (0.7) | 0.069 | 2.2 (0.7) | 2.3 (0.7) | 0.236 |
| Crossing infrastructure | 2.0 (0.7) | 2.2 (0.8) | 0.086 | 2.0 (0.7) | 2.1 (0.7) | 0.697 | 2.0 (0.7) | 2.1 (0.7) | 0.458 |
| Paths infrastructure | 2.5 (0.8) | 2.6 (0.9) | 0.379 | 2.5 (0.9) | 2.6 (0.8) | 0.379 | 2.5 (0.8) | 2.5 (0.9) | 0.894 |
| Walking and cycling infrastructure | 2.2 (0.5) | 2.2 (0.6) | 0.276 | 2.2 (0.6) | 2.2 (0.6) | 0.708 | 2.1(0.5) | 2.2(0.6) | 0.357 |
| Aesthetics | 2.7 (0.6) | 2.6 (0.7) | 0.099 | 2.7 (0.6) | 2.6 (0.6) | 0.237 | 2.6(0.7) | 2.7(0.6) | 0.417 |
| Crime safety | 2.9 (0.8) | 2.6 (0.8) | 0.012* | 2.7 (0.8) | 2.8 (0.8) | 0.722 | 2.6(0.8) | 2.8(0.8) | 0.022* |
| Traffic safety | 2.5 (0.7) | 2.6 (0.7) | 0.431 | 2.6 (0.7) | 2.5 (0.7) | 0.912 | 2.6(0.7) | 2.5(0.7) | 0.931 |
| Personal safety | 2.7 (0.6) | 2.8 (0.6) | 0.491 | 2.7 (0.6) | 2.8 (0.6) | 0.523 | 2.7(0.6) | 2.8(0.6) | 0.472 |
| Stranger danger | 2.2 (0.9) | 2.0 (0.9) | 0.067 | 2.5 (0.6) | 2.5 (0.5) | 0.386 | 2.1 (0.1) | 1.9 (0.1) | 0.194 |

PA; Physical Activity. MVPA, moderate-to-vigorous physical activity; SD, standard deviation; LSES, low socioeconomic status; HSES, socioeconomic status;

** p<0.001,

*p<0.05.

SES), higher parental perceptions of sidewalk infrastructure (β = -12.01, p <0.05) and walking/cycling infrastructure (β = -14.72, p <0.05) were associated with less MVPA. While none of the home environment variables were related to MVPA among girls, none of the home and neighborhood environment attributes were significantly related to MVPA among boys.

## Discussion

The current study investigated the associations between parental perceptions of home and neighborhood environment attributes and children's accelerometer-measured MVPA in a sample of Ugandan children. We found few and inconsistent associations of the home and neighborhood environments with children's MVPA, which replicated the patterns of evidence on this topic in LMICs [28]. To our knowledge, no study has been published on the relationship of the home and neighborhood environment on the PA behavior of children in Uganda.

This study's main finding was that one of five home environment attributes and two of 14 neighborhood environment attributes were significantly associated with children's MVPA in the full sample. However, two of the three significant associations were unexpected. At the home environment level, higher parental perception of play equipment at home was related to lower levels of children's MVPA. Similarly, the International Study of Childhood Obesity, Lifestyle, and the Environment found an inverse association between higher perception of play equipment at home and children's MVPA in Kenya (a low-income country), unlike the pattern

**Table 3. Results of full model, school type (school SES) and sex-stratified models assessing correlations between home and neighborhood environmental characteristics and children's MVPA.**

| Overall | | School type (school SES) | | Sex | | | |
|---|---|---|---|---|---|---|---|
| | | Private (HSES) | Public (LSES) | Boys | | Girls | |
| | | | | Private (HSES) | Public (LSES) | Private (HSES) | Public (LSES) |
| **Home Environment** | | | | | | | |
| Parental support for PA | 5.30 (0.240) | 3.50 (0.404) | 9.88 (0.176) | 9.78 (0.140) | -3.58 (0.807) | -2.55 (0.622) | 10.48 (0.174) |
| Parental rules for PA | -1.48 (0.076) | -0.79 (0.249) | -0.34 (0.814) | -0.69 (0.585) | 0.60 (0.855) | -0.972 (0.270) | -1.00 (0.493) |
| Media equipment in child's bedroom | -0.14 (0.934) | 0.46 (0.752) | -0.50 (0.870) | 0.77 (0.717) | 4.58 (0.353) | -0.35 (0.858) | -2.97 (0.398) |
| Child's personal media equipment | -3.37 (0.081) | 0.46 (0.782) | -2.05 (0.577) | 1.42 (0.570) | 1.42 (0.823) | -1.11 (0.617) | -4.39 (0.283) |
| Play equipment at home | -2.37 (< 0.001) ** | 0.23 (0.698) | -1.68 (0.187) | 0.74 (0.433) | -0.27 (0.917) | -0.44 (0.545) | -2.07 (0.119) |
| **Neighborhood Environment** | | | | | | | |
| Residential density | 2.70 (0.004) * | 1.13 (0.345) | -0.80 (0.550) | 2.04 (0.302) | -2.95 (0.191) | 0.82 (0.570) | 1.04 (0.494) |
| Land use mix-diversity (destinations) | -2.37 (0.279) | -1.24 (0.562) | -0.55 (0.870) | 1.72 (0.579) | 1.11 (0.865) | -5.45 (0.056) | 2.33 (0.530) |
| Land use mix-diversity (recreation) | -1.11 (0.487) | -0.93 (0.536) | -0.65 (0.801) | 0.07 (0.974) | 0.55 (0.901) | -2.08 (0.278) | -2.83 (0.326) |
| Land use mix-access | -4.24 (0.11) | -3.27 (0.176) | -6.86 (0.112) | 0.39 (0.916) | -1.94 (0.794) | -6.54 (0.031) * | -11.12 (0.026) * |
| Street connectivity | 2.87 (0.298) | -2.41 (0.372) | 2.11 (0.624) | -1.04 (0.826) | 13.63 (0.071) | -3.64 (0.234) | -3.50 (0.457) |
| Sidewalks infrastructure | -3.21 (0.168) | -0.48 (0.814) | -6.49 (0.108) | -0.44 (0.897) | -5.58 (0.359) | -1.06 (0.671) | -12.01 (0.015) * |
| Crossing infrastructure | -0.16 (0.942) | 0.49 (0.826) | -4.79 (0.161) | 6.37 (0.095) | -3.50 (0.510) | -3.20 (0.206) | -6.33 (0.120) |
| Paths infrastructure | -0.86 (0.652) | 0.52 (0.768) | -4.50 (0.144) | 1.92 (0.515) | -2.37 (0.603) | -0.35 (0.868) | -5.11 (0.181) |
| Walking and cycling infrastructure | -2.32 (0.423) | 0.27 (0.921) | -9.09 (0.050) | 3.89 (0.400) | -6.05 (0.374) | -2.28 (0.468) | -14.72 (0.010) * |
| Aesthetics | -2.54 (0.317) | 2.55 (0.285) | -4.13 (0.311) | 6.11 (0.106) | -5.32 (0.397) | -0.87 (0.766) | -6.17 (0.207) |
| Crime safety | -5.29 (0.006) * | -3.80 (0.038) * | -1.85 (0.562) | -3.61 (0.212) | 5.73 (0.290) | -3.71 (0.103) | -5.05 (0.154) |
| Traffic safety | 1.99 (0.375) | 0.12 (0.953) | 2.47 (0.506) | -0.52 (0.870) | 7.23 (0.229) | 0.963 (0.712) | -1.94 (0.650) |
| Personal safety | 0.09 (0.972) | 1.07 (0.649) | -3.53 (0.424) | 0.11 (0.976) | -1.69 (0.797) | 2.75 (0.348) | -6.85 (0.204) |
| Stranger danger | 2.84 (0.106) | -0.36 (0.825) | 3.33 (0.245) | 0.63 (0.788) | 1.85 (0.697) | -2.84 (0.221) | 0.78 (0.817) |

PA; Physical Activity. MVPA, moderate-to-vigorous physical activity; SD, standard deviation; LSES, low socioeconomic status; HSES, socioeconomic status;

** p <0.001,

* p <0.05.

in the HICs countries of Australia, Canada, and Finland, where more play equipment at home was related to more MVPA among children [41]. Perhaps, compared to children in HICs, the availability of play equipment at home may not be as important to children in LICs as having unrestricted access and time to use the equipment at home [42]. Social support is an important construct that can help explain children's PA behavior at home [13, 15, 43], but consistent with previous LIC studies [6, 34, 41], we did not find associations between social support and children's MVPA in the present study. Social support may be more important to children from HICs where children engage in more organized sports activities that require parents to pay for membership and transport their children to activity venues [41], unlike free play and transportation PA, which is predominant in LICs like Uganda [11].

At the neighborhood environment level, higher parental perception of safety from crime was unexpectedly related to lower levels of children's MVPA, suggesting that higher parental perception of insecurity from crime leads to engagement in more MVPA among the children. The usually reported unexpected and inverse associations of crime safety with PA behaviors of children and adults in Africa is largely a reflection of environmental injustice, under conditions in which people have no choice but to walk for transportation to destinations and utilitarian purposes regardless of the pervasive insecurity from crime and traffic [44]. To address the unintended consequences of the circumstances in which children will have to be active

despite the high perception of crime in the neighborhood, there is a recent call for "physical activity security" as an agenda for creating an enabling PA environment in LMICs and other highly inequitable settings [45].

The only correlation of children's MVPA in the expected direction was the perception of higher residential density, replicating findings from international studies [6, 46]. High residential density neighborhoods have access to various destinations (friends' homes, shops, or transit stops) and are well connected, providing more opportunities for children to walk, thus increasing PA [47, 48]. High residential density has also been associated with adolescent PA [49, 50] and adult PA [48], so this construct could be an important component of neighborhood walkability that is applicable across the age spectrum, even in Africa [51].

Sex-specific analyses provided inconsistent patterns of findings. Irrespective of school type, girls unexpectedly engaged in less MVPA when their parents perceived more access to various destinations, services, and transit stops. A review study also reported negative associations between access to various destinations and PA among children [25]. Girls attending public (low SES) schools engaged in less MVPA if their parents perceived more walking and cycling infrastructure, particularly sidewalks. Because public (low SES) schoolgirls are more likely to live in areas that are more compact with high residential density, this diminishes space for walking and cycling, reducing their PA [52]. For example, the description of the neighborhood environment characteristics in our study "S1 Table" revealed that most of the parents perceived the sidewalks to be indistinct from the road (vehicle traffic) and were not well maintained. Parents also perceived more informal walking or foot paths with no designated places to walk or cycle. These conditions of walking and cycling infrastructure may discourage girls from walking, thereby decreasing their PA.

Our findings raise questions about why girls participated in less MVPA despite parental perceptions of the presence of favorable neighborhood environment attributes. Cultural, social, and gender norms coupled with parental rules and concerns about unsupervised PA in the neighborhood may be critical factors. Cultural and social norms may initiate unfavorable gender stereotypes that limit girls' social and behavioral expressions; for example, girls and boys may be offered distinct activities based on their gender, most commonly encouraging boys to play vigorously and girls to play quietly [53]. Literature also shows significant differences in independent mobility; boys experience far more freedom and spend more time in activity-enhancing environments than girls, particularly outdoors [46, 54, 55]. The time that children spend outdoors is consistently and positively correlated with their PA [56]. Other leisure time, particularly sedentary activities, such as engaging in screen-based activities (television, videogames, and computer use), may attract girls to stay at home rather than go out into the neighborhood [57]. On the other hand, the lack of association between parental-perceived neighborhood attributes and MVPA in boys may be due to the definition of the neighborhood in our study (the area surrounding the child's home that they could walk to within 10 to 15 minutes). Since boys are reported to have a higher independent mobility, it is possible that environmental attributes outside the neighborhood may be associated with their MVPA [46]. In subsequent studies, it would be beneficial to examine the facilitators and barriers that girls encounter to achieve sufficient PA at home and in their neighborhoods.

## Strength and limitations

The primary strength of this study is the use of accelerometers to measure children's MVPA in an understudied region. This study also provides preliminary evidence on the link between home and neighborhood environment attributes and PA behaviors of children in Uganda. However, the results of the current study should be interpreted with caution because of its

cross-sectional study design, which limits the ability to make causal inferences. In addition, both the home and neighborhood environment characteristics were self-reported and, as such, may have been vulnerable to social desirability and bias. The low response rate and recruitment of participants from Kampala city limits the generalizability of the findings to other populations. The current study did not assess specific types of PA, such as leisure and transportation [58], and the home and neighborhood context of the MVPA outcomes [30, 34]. We also combined different items for the home and neighborhood environment variables into one scale, unlike other studies in which individual items were assessed [16, 33], which might have obscured the associations of the home and neighborhood environment with children's MVPA. Previous studies have shown that the home environment is more likely to be correlated with sedentary behavior, which occurs predominantly at home, than with MVPA, which accumulates across the entire day in different settings [15, 59]. In addition, there was limited variability in parental perceptions of the home and neighborhood environments in the current study, which may have reduced the power to detect associations with children's MVPA [60]. Therefore, the correlation of the home and neighborhood environment with children's PA needs to be studied further with increased specificity and improved measures to improve the quality of evidence.

## Conclusion

Overall, very few home and neighborhood environment attributes were supported as correlates of MVPA among Ugandan children. The patterns of influence of the home and neighborhood environments on Ugandan children's PA may be gender specific and different from those in the HICs. Future qualitative studies exploring the perceptions, facilitators, and barriers to PA participation among girls in LMICs are important since girls exhibit lower levels of PA than boys. More research is needed on correlations between the built environment and PA in Uganda, particularly using objective measures of the built environment and longitudinal cohort studies to better guide effective health promotion interventions and policies.

## Supporting information

**S1 Table. Description of parent perceptions of home and neighborhood environment attributes.**
(DOCX)

**S1 File. Parent/guardian questionnaire.**
(DOCX)

**S2 File. Data set.**
(PDF)

## Acknowledgments

The authors appreciate the Directorate of Education and Social Services Kampala Capital City Authority (KCCA) for supporting them in accessing schools. We appreciate the research assistants who greatly contributed to data collection. We are also grateful to all school head teachers, teachers, parents/guardians, and the children who participated in this study. We thank the Physical Activity and Health Laboratory at the University of Massachusetts Amherst, USA, for their support in accelerometry data management and interpretation.

## Author Contributions

**Conceptualization:** Bernadette Nakabazzi, Lucy-Joy M. Wachira, Adewale L. Oyeyemi, Vincent O. Onywera.

**Formal analysis:** Ronald Ssenyonga.

**Funding acquisition:** Bernadette Nakabazzi.

**Investigation:** Bernadette Nakabazzi.

**Methodology:** Bernadette Nakabazzi, Lucy-Joy M. Wachira, Adewale L. Oyeyemi, Vincent O. Onywera.

**Software:** Ronald Ssenyonga.

**Supervision:** Lucy-Joy M. Wachira, Adewale L. Oyeyemi, Vincent O. Onywera.

**Visualization:** Bernadette Nakabazzi.

**Writing – original draft:** Bernadette Nakabazzi, Lucy-Joy M. Wachira, Adewale L. Oyeyemi, Vincent O. Onywera.

**Writing – review & editing:** Bernadette Nakabazzi, Lucy-Joy M. Wachira, Adewale L. Oyeyemi, Ronald Ssenyonga, Vincent O. Onywera.

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
