## [Decision Letter · Decision Letter 0]

15 Aug 2021

 PGPH-D-21-00222 Parental perceived home and neighbourhood environmental correlates of accelerometer-measured physical activity among school-going children in Uganda. PLOS Global Public Health

Dear Dr. Nakabazzi,

Thank you for submitting your manuscript to PLOS Global Public Health. After careful consideration, we feel that it has merit but does not fully meet PLOS Global Public Health’s publication criteria as it currently stands. Therefore, we invite you to submit a revised version of the manuscript that addresses the points raised during the review process.

We look forward to receiving your revised manuscript.

Kind regards,

Jenil Patel, MBBS, MPH, PhD

Academic Editor

Journal Requirements:

Additional Editor Comments (if provided):

Reviewers' comments:

Reviewer's Responses to Questions

**Comments to the Author**

1. Does this manuscript meet PLOS Global Public Health’s publication criteria? Is the manuscript technically sound, and do the data support the conclusions? The manuscript must describe methodologically and ethically rigorous research with conclusions that are appropriately drawn based on the data presented.

Reviewer #1: Yes

Reviewer #2: Yes

Reviewer #3: Yes

2. Has the statistical analysis been performed appropriately and rigorously?

Reviewer #1: Yes

Reviewer #2: Yes

Reviewer #3: I don't know

3. Have the authors made all data underlying the findings in their manuscript fully available (please refer to the Data Availability Statement at the start of the manuscript PDF file)?

Reviewer #1: Yes

Reviewer #2: Yes

Reviewer #3: Yes

4. Is the manuscript presented in an intelligible fashion and written in standard English?

Reviewer #1: Yes

Reviewer #2: Yes

Reviewer #3: Yes

5. Review Comments to the Author

Reviewer #1: This paper presents a cross sectional study on built environment correlates of objectively-measured physical activity among children in Uganda. The paper is very well written and would be useful to public health professionals and policy makers. The findings will help inform the development of effective physical activity interventions and policies. Congratulations! I am offering some suggestions for consideration before publication.

Abstract

1- Given the scope of the journal and the physical inactivity issue, I would recommend starting the abstract with a sentence highlighting the significance of physical inactivity problem and how it is related to non-communicable diseases.

Introduction

Excellent introduction! Some minor comments below for improvement.

2. Line 50: please define the acronym PA.

3. Line 53: please define the acronym MVPA

4. Please define the MVPA (i.e., using METs)

5. I would recommend including some more supporting statistics about the issue and the burden associated with it (e.g., healthcare or societal costs of physical inactivity).

6. The authors used objective measure for PA (i.e., accelerometer), which is excellent. I would recommend that you highlight the current lack of objectively-measured PA research in LMIC compared to self-reported (which is subject to reporting bias) and that this study addresses this gap.

Materials and methods

7. I suggest that the authors include a section for study setting/location to provide a brief description of the place/area/neighbourhood where this study was conducted.

8. May I ask about the approach you adopted to recruit the sample?

Discussions

9. A unique finding in this study is that the influence of the home & neighbourhood environments on Ugandan children’s PA may be gender-specific. I suggest reflecting more on this by, for example, discussing the concept of embodiment which is relevant here (i.e., how Ugandan girls embody the social environment where they live and the associated gendered norms that restrict their PA participation). This is a major issue that extends to Western countries and can actually explain the inequities and low PA participation rates among immigrant girls (from LMICs) compared to boys in Western countries.

Conclusion

10. Line 351: Please correct to “environments on Ugandan children’s PA”.

11. I would suggest that the authors add additional recommendation highlighting the need for qualitative studies to understand the perceptions and barriers to PA engagement among girls in LMICs. This can provide more insight into the required needs to best design equitable interventions.

Reviewer #2: The study analyzes the Uganda context and aims to explore the relationship between home and neighborhood environment (as perceived by parents) and moderate to vigorous physical activity. The manuscript is relevant, well-written, and scientifically sound. The topic fits within the scope of the Journal.

I consider the introduction is clear and covers a general background. An apparent gap has been identified in this section. However, I believe it would be more relevant if the authors could include more recent work (papers published in 2017-2021) since it could increase the interest of potential readers.

Since the research is highly focused on Kampala (Uganda) population, I consider the authors should include a description of the characteristics of the city and its environment. I would help readers out of Uganda to better understand the research results. The sample is a limitation of the study and the authors have stated it in the limitations section. For that reason, the sample characteristics explanation gains importance.

In the methods section, the authors clearly state the eligibility criteria. Were there any exclusion criteria? Please, include them, if applicable.

In my view, the results and the discussion are clear and understandable. However, in Table 3, in file “play equipment at home”, the “overall” result shows p>0.001. I wonder if this is a typo error and should be p<0.001 instead. Please confirm this and correct it, if applicable.

The conclusions could emphasize the practical implications of the study. This way, the article would be potentially meaningful for professionals in the field, even policymakers.

The effort made by the authors is appreciated and the comments are expected to be useful.

Reviewer #3: Please, find some comments and suggestions for the manuscript entitled: “Parental perceived home and neighborhood environmental correlates of accelerometer-measured physical activity among school-going children in Uganda”. I would like to congratulate Authors for this interesting study. Nevertheless, I consider there are some small changes that should be done.

Line 98: The abbreviation of the PA has been used in the manuscript earlier. Please, use only the abbreviation.

Line 102: The abbreviation of the LICs has been used in the manuscript earlier. Please, use only the abbreviation.

Line 192: Why do you think that the type of school may differentiate the results (private vs. public school)? How were the schools selected based on SES? Could you describe that process?

Line 201 (in general all tables): I suggest removing the horizontal lines in tables (or even only some of them), however, it should stick to the Journal style (it refers to all tables in the manuscript).

Line 256: The abbreviation of the PA has been used in the manuscript earlier. Please, use only the abbreviation.

Line 353: The same as above.

6. PLOS authors have the option to publish the peer review history of their article (what does this mean?). If published, this will include your full peer review and any attached files.

**Do you want your identity to be public for this peer review?** For information about this choice, including consent withdrawal, please see our Privacy Policy.

Reviewer #1: No

Reviewer #2: No

Reviewer #3: No

---

## [Editor Report · Decision Letter 1]

15 Nov 2021

Parental perceived home and neighbourhood environmental correlates of accelerometer-measured physical activity among school-going children in Uganda.

PGPH-D-21-00222R1

Dear Dr. Nakabazzi,

We're pleased to inform you that your manuscript has been judged scientifically suitable for publication and will be formally accepted for publication once it meets all outstanding technical requirements.

Within one week, you'll receive an e-mail detailing the required amendments. When these have been addressed, you'll receive a formal acceptance letter and your manuscript will be scheduled for publication.

An invoice for payment will follow shortly after the formal acceptance. To ensure an efficient process, please log into Editorial Manager at https://www.editorialmanager.com/pgph/ click the 'Update My Information' link at the top of the page, and double check that your user information is up-to-date. If you have any billing related questions, please contact our Author Billing department directly at authorbilling@plos.org.

Kind regards,

Jenil Patel, MBBS, MPH, PhD

Academic Editor